# Comparison of orbital structures between age-related distance esotropia and acute acquired concomitant esotropia

Jin Hee Kim[1], Won-Jin Moon[1,2], Hyun Jin Shin[1,3,4,5]*

1 School of Medicine, Konkuk University, Chungju, Republic of Korea, 2 Department of Radiology, Konkuk University Medical Center, Konkuk University School of Medicine, Seoul, Republic of Korea, 3 Department of Ophthalmology, Konkuk University Medical Center, Seoul, Republic of Korea, 4 Research Institute of Medical Science, Konkuk University School of Medicine, Seoul, Republic of Korea, 5 Institute of Biomedical Science & Technology, Konkuk University, Seoul, Republic of Korea

* shineye@kuh.ac.kr

## Abstract

### Purpose

To compare orbital magnetic resonance imaging (MRI) findings of age-related distance esotropia (ARDE) with those of age-matched healthy controls and acute acquired concomitant esotropia (AACE) patients.

### Methods

Coronal MRI scans of 16 ARDE cases, 15 AACE cases, 26 elderly controls cases, and 22 young controls cases were analyzed to measure (1) the displacement angles of the lateral rectus (LR) and medial rectus (MR), (2) the LR tilting angle, (3) the ratio of the MR-to-LR cross-sectional areas (CSAs), and (4) the superior rectus (SR) downward displacement ratio (SDR).

### Results

ARDE patients showed significant LR sagging relative to elderly controls, by 5.1° ($p = 0.048$), while there was no significant LR displacement in AACE relative to young controls. ARDE patients also exhibited marked LR tilting relative to all other groups. Additionally, the MR-to-LR CSA ratios were 26% ($p = 0.002$) and 27% ($p = 0.001$) higher in the ARDE and AACE groups than in the controls, respectively, indicating imbalance in the horizontal rectus (HR). Additionally, SDR values were higher in ARDE and elderly controls, suggesting that a closer proximity of the SR muscle to the globe is an age-related alteration.

**Data availability statement:** All relevant data are within the manuscript and its Supporting Information files.

**Funding:** This research was supported by Basic Science Research Program through the National Research Foundation of Korea (NRF) grant funded by the Korea government (MSIT) (No. RS-2023-00251281). The funders had no role in study design, data collection and analysis, decision to publish, or preparation of the manuscript.

**Competing interests:** The authors have declared that no competing interests exist.

## Conclusion

The findings demonstrate that ARDE is characterized by unique orbital changes, particularly in the LR, that distinguish it from AACE. The observed increase in the MR-to-LR CSA ratio among the esotropia groups points to an imbalance of HR tension. These MRI-based insights enhance our comprehension of the unique pathophysiology of these conditions, helping to distinguish ARDE from AACE in acquired esotropia and guiding customized treatments strategies.

---

## 1. Introduction

Strabismus can stem from various causes that are broadly categorized into anatomical and innervational factors [1,2]. Anatomical factors pertain to structural abnormalities within the extraocular muscles (EOM) or the orbital anatomy [3,4], whereas innervational factors are related to disruptions in the neurological control of eye movements. The coordination of eye movements is dependent on precise signals from the brain to the eye muscles, with any interference of this signaling potentially resulting in misalignment [5,6].

The introduction of high-resolution magnetic resonance imaging (MRI) has shed light on the anatomical etiologies of strabismus [7,8]. Notably, Chaudhuri & Demer [4] identified the phenomenon of sagging-eye syndrome or age-related distance esotropia (ARDE), which is attributed to the dislocation of orbital pulleys and alterations in the positioning of EOM in the elderly. That study yielded MRI evidence of ARDE, showcasing superotemporal bowing of the lateral rectus (LR)–superior rectus (SR) complex. It has been hypothesized that age-associated deterioration of this complex could cause downward displacement of the LR pulley, leading to esotropia, potentially accompanied by vertical misalignment.

Acute acquired concomitant esotropia (AACE) is marked by the sudden onset of esotropia and double vision in older children and adults [9]. The prevalence of AACE is thought to be increasing, with recent reports suggesting a link to the increasing prevalence of performing near-vision tasks and the usage of digital devices including smartphones [10,11], notably during the COVID-19 lockdown. This increase in AACE cases is becoming a significant public health concern that demands detailed exploration. Although the exact pathophysiology of AACE is unknown, some EOM abnormalities causing imbalance in the horizontal rectus (HR) have been reported. Hayashi et al. reported that the medial rectus (MR) insertion widths were larger in patients with AACE than in a control group undergoing retinal detachment surgery [12]. Additionally, Chen et al. used MRI to observe changes in the size and volume of the EOM in AACE patients [13].

Despite their different typical onset ages, ARDE and AACE share several similarities, including sudden-onset concomitant esotropia. Recent studies have explored the alterations in the rectus pulleys associated with ARDE, but there has been a lack of studies analyzing the mechanical aspects of AACE such as pulley position and EOM size. In the current study we compared these two esotropia groups to

determine if any features of AACE are shared with ARDE, and to identify the presence of any anatomical abnormalities in AACE. This study performed a thorough comparison of orbital MRI findings in AACE with those in an age-matched healthy group and an ARDE group.

## 2. Materials and methods

This retrospective case–control study was conducted at the Department of Ophthalmology, Konkuk University Medical Center, Seoul, South Korea. The study was performed in accordance with the principles of the Declaration of Helsinki, and its protocol was approved by the institutional review board and ethics committee of Konkuk University Medical Center (approval number: 2024-11-046). The data used in this study were accessed for research purpose in 07/12/2024. We reviewed the medical records of patients diagnosed with AACE and ARDE who underwent high-resolution orbital MRI between 1 March 2021 and 30 December 2022. The authors had access to information that could identify individual participants during data collection. These records were compared with those of age-matched controls who underwent MRI for routine health examinations during the same time period. The need to obtain informed consents from the participants was waived by the ethics committee.

The inclusion criteria for ARDE included no history of strabismus, normal abduction and adduction, onset of symptoms after 60 years of age, concomitant esotropia during lateral gaze, symptom of diplopia at distance, and distance esotropia exceeding near esotropia by at least 5 prism diopters [14]. The inclusion criteria for AACE included esotropia with sudden-onset diplopia that developed after 18 years of age and the angle of deviation in near and distance, lateral gaze showed a difference within 5 prisms diopter, with no limitation of eye movement [15]. The exclusion criteria were as follows: (1) history of ocular or head trauma, (2) history of ocular surgery (except for refractive surgery), (3) cerebrovascular or central nervous system disorders, (4) systemic disorder affecting the eye and EOM such as thyroid eye disease, myasthenia gravis, or IgG4-related disease, (5) other causes of binocular diplopia or heterotopia such as superior oblique palsy, abducens palsy, or heavy eye syndrome, (6) accommodative spasm or accommodative esotropia, or (7) eyes with spherical equivalents of >3.00 or <−4.00 diopters.

### 2.1. Orbital MRI analysis

MRI scans were performed on a SIEMENS Vida 3T with 20 head/neck coil. Coronal T1-FLAIR imaging of the orbits was performed with the following parameters: TR/TE = 2690/12 ms, FOV = 140 mm, flip angle = 150° slice thickness 3 mm with no gap, matrix = 256 × 256 pixels, and NEX = 1, in plane resolution 0.273 × 0.273 mm. Each scan was performed with the subject's head stabilized and primary gaze maintained, with the subject instructed to avoid unnecessary movements. Orbital MRI scans of the coronal plane at 3 mm anterior to the globe–optic-nerve junction were processed using ImageJ image analysis software (US National Institutes of Health) [16,17].

Each rectus muscle and the globe were traced using the ImageJ wand tool that automatically selects threshold images of these structures created by the *Image>Adjust>Threshold* tool (Kim JH). These selected images were used as references for freehand drawings to make the outlining of structures more accurate. The geometric center point of each rectus muscle was found using the *Area Centroid* measuring function in ImageJ, while its cross-sectional area (CSA) was found using the *Area* measuring function. The horizontal line passing through the centroids of the two eyeballs was defined as a horizontal reference (HR) line, and a line perpendicular to the HR line passing through each centroid of the eyeball was defined as a vertical reference (VR) line (Fig. 1A and 1B). MRI imaging measurements were performed by the researcher (Kim JH) and confirmed by the supervisor (Shin HJ).

### 2.2. Outcome measurements

The angle of deviation was quantified using a prism alternate-cover test conducted at 4 m for distance viewing and 33 cm for near viewing. The following variables were defined and assessed (Fig 1):

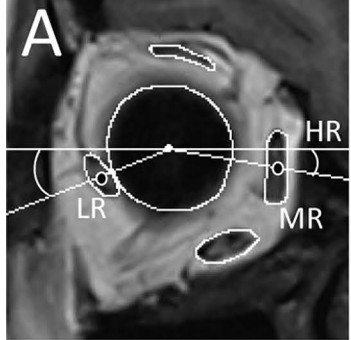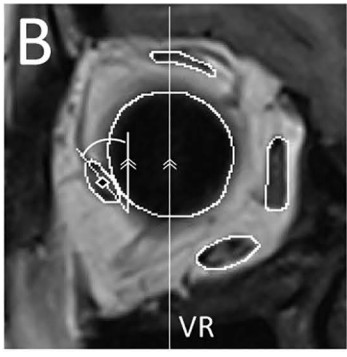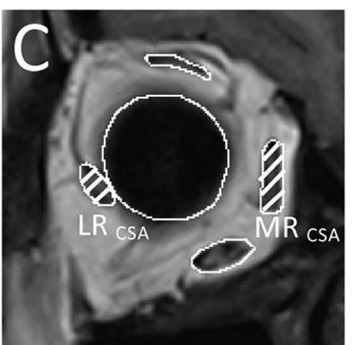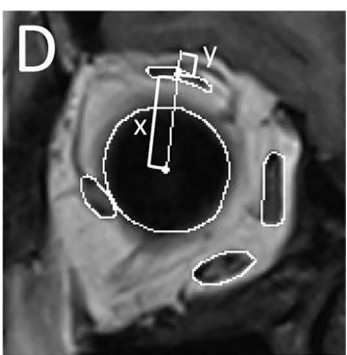

**Fig 1. Methodology for measuring variables using T1-weighted standard coronal plane magnetic resonance images, captured 3 mm anterior to the junction between the globe and optic nerve. (A)** Measurement of horizontal rectus (HR) displacement angles (●) relative to a horizontal reference line. **(B)** Measurement of the lateral rectus (LR) tilt angle (♦) relative to a vertical reference line. **(C)** Measurement of the ratio of the cross-sectional areas (CSAs) of the LR and the medial rectus (MR), where the CSA of each muscle is indicated by slanted lines for visual clarity. **(D)** Measurement of the superior rectus (SR) downward displacement ratio, calculated as the distance from the SR centroid to the bony orbit (*y*) divided by the distance from the globe centroid to the SR centroid (*x*).

1. HR displacement angle, which is the angle formed by the HR relative to the HR line. Positive and negative displacement angles for the LR and MR indicate positions above and below the HR line, respectively.

2. LR tilting angle, which is the angle between the major axis of the LR and a line parallel to the VR line. Positive and negative tilting angles correspond to temporal and nasal tilts of the LR, respectively [17].

3. CSAs of the MR and LR, as well as their ratio, were measured to assess the balance between the two muscles. A higher MR-to-LR CSA ratio indicates that the medial rectus has a relatively larger cross-sectional area compared to the lateral rectus.

4. SR downward displacement ratio (SDR) between the distance from the SR centroid to the bony orbit, and the distance from the globe centroid to the SR centroid. A line is drawn from the globe centroid through the SR centroid extending to the orbital wall, and the partition ratio of the line divided by the SR centroid is computed. Increased SDR indicates that the SR is positioned closer to the globe.

The calculated variables were analyzed statistically using standard software (version 20.0, SPSS for Windows, SPSS, Chicago, IL, USA). The Shapiro-Wilk test was used to determine whether data conformed to a parametric (Gaussian) or nonparametric (non-Gaussian) distribution. One-way analysis of variance and the Bonferroni post-hoc test were used to analyze group differences. Significance was defined as $p < 0.05$.

## 3. Results

Sixteen orbits of ARDE patients (mean age 71.3 ± 10.4 years, ranging from 60 to 89 years) and 15 orbits of AACE patients (mean age 28.4 ± 6.6 years, ranging from 22 to 36 years) were included in the study. Age-matched controls consisted of 22 orbits from younger participants (mean age 26.2 ± 7.8 years, ranging from 18 to 34 years) and 26 orbits from older participants (mean age 72.8 ± 6.4 years, ranging from 66 to 88 years). The baseline clinical characteristics and the demographics of the participants are outlined in Table 1.

### 3.1. HR displacement angles

The LR showed greater downward displacement in the elderly control group than in the young control group. Specifically, individuals with ARDE demonstrated notable LR sagging, with a mean deviation of 5.3° relative to the elderly control

**Table 1. Baseline characteristics and MRI measurements in each group.**

| | ARDE | Elderly control | AACE | Young control | P value | | |
| --- | --- | --- | --- | --- | --- | --- | --- |
| | | | | | ARDE vs elderly control | AACE vs young control | ARDE vs AACE |
| **N (= no. of orbits)** | 16 | 26 | 15 | 22 | | | |
| **Mean age (year)** | 71.3±10.4 | 72.8±6.4 | 28.4±6.6 | 26.2±7.8 | | | |
| **Sex (male: female)** | 6: 3 | 7: 6 | 3: 5 | 5: 7 | | | |
| **Mean angle of far deviation (PD)** | 6.9±3.6 | N/A | 18.1±10.2 | N/A | | | |
| **Mean angle of near deviation (PD)** | 3.6±3.6 | N/A | 16±11.6 | N/A | | | |
| **HR displacement angle** | | | | | | | |
| Mean MR displacement angle | -7.3±6.0 | -3.9±9.2 | −4.1±6.4 | -3.6±3.6 | 0.13 | 0.78 | 0.12 |
| Mean LR displacement angle | -13.4±4.7 | -8.1±8.1 | -3.3±5.7 | -0.98±5.9 | 0.048* | 0.25 | 0.001** |
| **Mean LR tilting angle (degree)** | 20.3±8.2 | 15.6±7.1 | -2.0±6.2 | -1.45±6.0 | 0.053 | 0.10 | 0.001** |
| **Mean SDR** | 0.34±0.1 | 0.35±0.1 | 0.29±0.1 | 0.28±0.1 | 0.63 | 0.70 | 0.28 |
| **Mean (MR/LR)$_{CSA}$** | 2.20±0.70 | 1.63±0.44 | 1.54±0.45 | 1.12±0.20 | 0.002** | 0.001*** | 0.005** |

Values with asterisk *, **, ***, are statistically different at probability values of $p \leq 0.05, \leq 0.01$ and $\leq 0.001$, respectively

LR, lateral rectus; MR, medial rectus; SR, superior rectus; ARDE, age-related distance esotropia; AACE, acute acquired concomitant esotropia; PD, prism diopters; CSA, cross-sectional area; SDR, superior rectus downward displacement ratio; ±indicates standard deviation

($p=0.048$). However, LR displacement did not significantly differ between AACE and young controls (Fig 2). MR displacement angles in both ARDE and AACE groups were comparable to their age-matched controls.

### 3.2. LR tilting angle

In the ARDE group, the mean LR tilting angle was 4.7° larger than that in the elderly control group, approaching statistical significance ($p=0.053$). No significant difference was found between AACE and young controls. However, ARDE patients showed a markedly greater LR tilting angle, 19.1° larger than AACE ($p=0.001$). The elderly control group also had significantly larger LR tilting angles than both AACE and young controls ($p<0.001$) (Fig 3).

### 3.3. MR-to-LR CSA ratio

In the ARDE group, the mean MR-to-LR CSA ratio was 26% higher than in elderly controls ($p=0.002$). Similarly, AACE patients had a 27% higher ratio than young controls ($p=0.001$) (Fig 4). Both esotropia groups showed elevated MR-to-LR CSA ratios compared to their age-matched controls.

### 3.4. SR downward displacement ratio

The SDR quantifies SR positioning relative to the globe and bony orbit. It was 18% higher in ARDE and 20% higher in older controls than in young controls. These findings suggest that the SR is more prone to downward displacement as age progresses (Fig 5).

## 4. Discussion

The aim of the study is to determine whether AACE shares features with ARDE and to identify anatomical abnormalities in these esotropia groups compared to age-matched controls. In the AACE group, the rectus muscles displayed normal morphology in terms of their displacements along the HR and VR lines, which indicated the typical pulley positions. Conversely, the orbital structures in the ARDE group notably differed from those in the AACE group, with the LR in the ARDE group being substantially more inferiorly displaced and tilted, due to degeneration of the LR-SR band. Both esotropia groups had a higher MR-to-LR CSA ratio than controls, suggesting an anatomical imbalance contributing to esotropia.

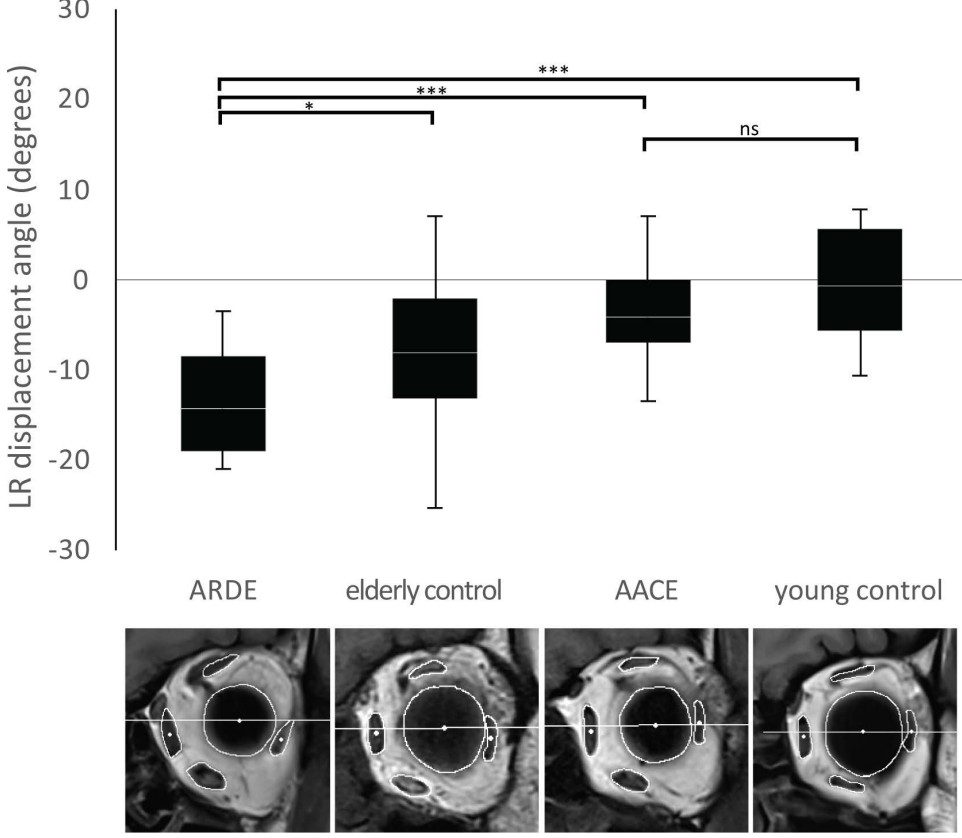

**Fig 2. Box plots indicating the distributions of the LR displacement angles across the four study groups, for the left orbits.** ARDE, age-related distance esotropia; AACE, acute acquired concomitant esotropia. $N_{ARDE} = 16$, $N_{elderly\ controls} = 26$, $N_{AACE} = 15$, $N_{young\ controls} = 22$. The top of each box represents the 75th percentile, the bottom of each box represents the 25th percentile, the line in the middle of each box represents the 50th percentile, and the whiskers represent the highest and lowest values. The ARDE group exhibits a significant inferior displacement of the LR, indicating a more-pronounced sagging relative to the other groups. *, $p < 0.05$; ***, $p < 0.001$; ns, not significant.

Additionally, the SR was positioned closer to the globe and farther from the orbital wall in ARDE patients and elderly controls, reflecting age-related orbital structure changes (S1 Fig).

Consistent with the prevailing understanding that ARDE is associated more strongly with mechanical alterations than with neurological issues [4], our findings revealed displacements in the EOM in the ARDE group. The connective-tissue pulleys of the rectus muscles are composed of collagen, elastin, and smooth muscle, and they undergo degeneration with age that leads to a mechanical relaxation of the HR pulleys [18,19]. Pulleys are known to play crucial roles in modulating the horizontal, vertical, and torsional actions of the rectus muscles [20,21]. Our observation of the LR sagging downwards by $-13.4 \pm 4.7°$ is consistent with Kawai et al. reporting a significant downward deviation of the LR ($-13.3 \pm 10.9°$) in individuals with ARDE relative to normal elderly subjects [22], further substantiating the mechanical basis of this condition.

In addition to sagging, the LR in the ARDE group was found to be obliquely tilted at $20.3 \pm 8.2°$, compared with a reference value of $24.1 \pm 9.5°$ observed in previous studies [22]. This tilting, alongside the sagging of the LR, was a prevalent characteristic in the elderly, rather than being seen exclusively in the ARDE group, although the degree of deviation was substantially more pronounced in ARDE cases. The angulation of the LR may stem from the pressure exerted by superotemporally migrating orbital fat, which is not restrained by the degenerated LR-SR band and its surrounding connective tissues [4]. Koornneef et al. reported that the intermuscular septum is thinnest in the superotemporal quadrant [23], making the LR-SR band particularly susceptible to age-related changes, consequently leading to tilting of the LR. This sagging

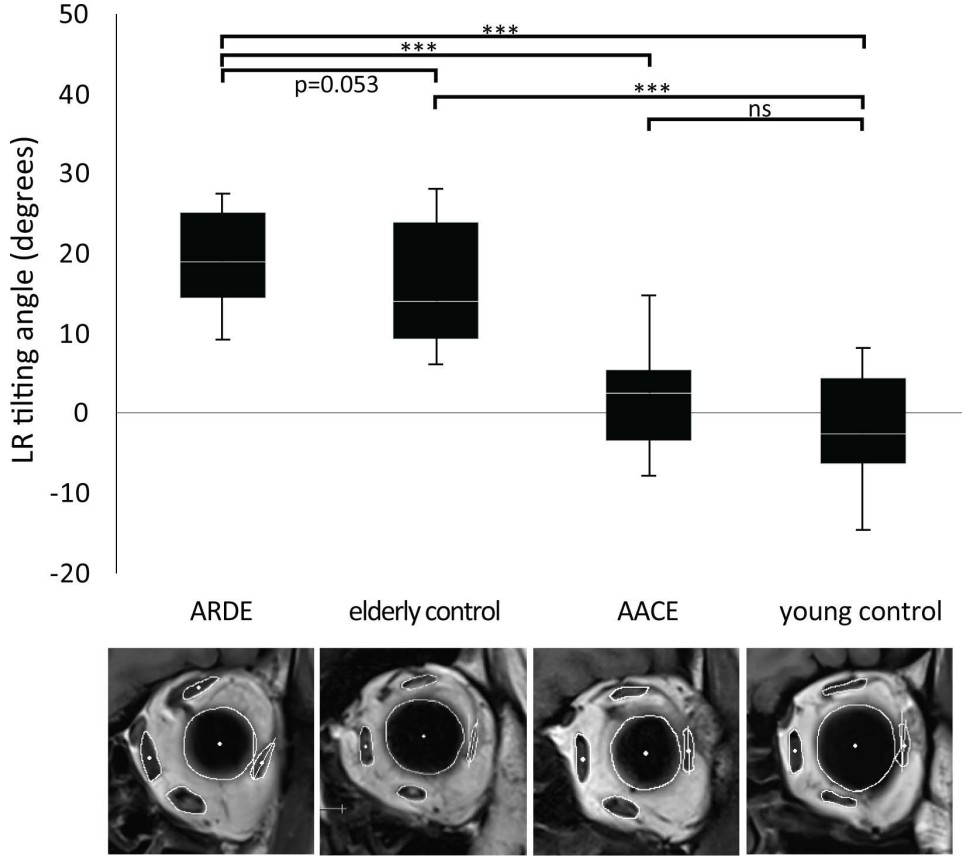

**Fig 3. Box plots indicating the distributions of the LR tilting angles across the four study groups.** The LR tilting correlates with aging and is notably more marked in the ARDE group. ***, $p < 0.001$.

and tilting of the LR impair its abduction capability, thus compromising the eye's divergence abilities and predisposing ARDE patients to esotropia during distance fixation.

This study also examined the MR-to-LR CSA ratio, which was significantly higher in both esotropia groups (ARDE and AACE) than in age-matched controls. These findings support the hypothesis that the MR becomes hypercontractile in concomitant esotropia, possibly due to increases in size and/or strength, while the size and strength of the LR might decrease, resulting in reduced abduction forces [24]. Altick et al. observed a reduction in the expression of genes related to contractility and an increase in the expression of genes associated with the extracellular matrix in the EOM of strabismic individuals [25], indicating a structural imbalance in these muscles. Previous research has also shown that the CSA of the MR is significantly larger in patients with concomitant esotropia than in normal controls, while this was not observed for the LR [24]. Our analysis suggests that assessing the MR-to-LR CSA ratio may provide insights into adduction-abduction imbalance, complementing the comparisons of the absolute sizes of these muscles.

In addition, our findings for the CSAs of the HR in the normal control group are consistent with previous normative orbital measurements in the Asian population [26]. The literature indicates that the CSA of the LR decreases with age, a trend that was also evident in our study. This decrease in CSA is consistent with the observation of age-related LR sagging, since a lax muscle with an elongated path is likely to have a reduced CSA, explaining why the MR-to-LR CSA ratio was higher in the elderly controls than the young controls. This age factor attributes the highest ratio observed in the ARDE group to sagging of the LR and hence muscle-path elongation being most prominent in ARDE.

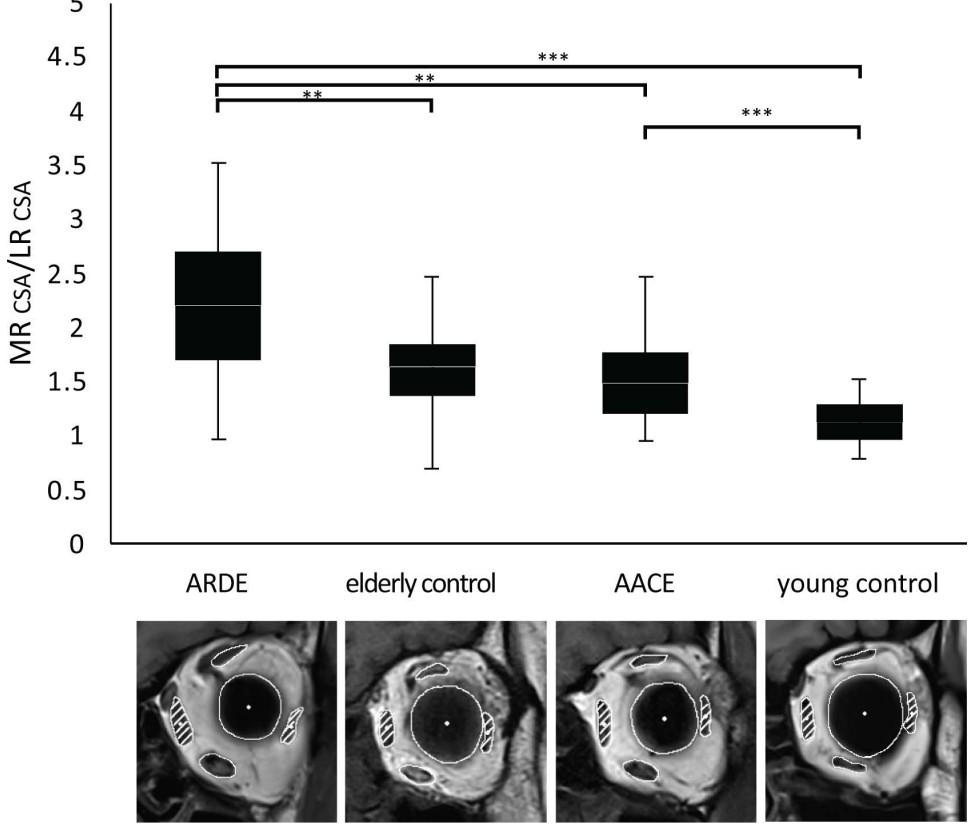

**Fig 4. Box plots indicating the relative sizes of the MR and LR across the four study groups.** The CSAs of both the MR and LR are indicated by slanted lines for clarity. This visualization highlights that the MR is notably larger than the LR, demonstrating a more-substantial imbalance in subjects with esotropia in contrast with their respective age-matched normal controls. **, $p < 0.01$; ***, $p < 0.001$.

In the AACE group, the ratio was relatively high despite the presence of an intact pulley system, suggesting that increased MR contractility [24] can lead to HR imbalances. These findings imply that the development of AACE is influenced by both neurological and mechanical factors. Previous studies have shown that short-acting topical cycloplegics can be effective for AACE patients [27]. They noted that reductions in esotropia angles were inversely related to the length of untreated esotropia, emphasizing the importance of early intervention. Further investigation is required to determine if the increased MR-to-LR CSA ratio in AACE is associated with a longer duration of strabismus, which would help clarify the link between anatomical changes of EOM and the duration of strabismus.

In older participants, including ARDE patients and elderly controls, the SR was positioned closer to the globe, as indicated by a significantly larger SDR. This aligns with Gomez et al., who reported geometrically closer SR to the globe in ARDE patients than younger controls [16]. The SR runs parallel to the orbital roof until reaching a connective-tissue pulley located just posterior to the globe's equator, where it curves along the globe to its insertion point [28]. In older age groups, a higher CSA ratio is observed, likely due to sagging and elongation of the LR muscle and suggests age-related weakening of connective tissues securing the SR. This structural weakening brings the SR closer to the globe and is accompanied by the inferior displacement of the LR pulley, illustrating the interplay of orbital changes with aging. These mechanical alterations may explain reduced saccadic velocity and impaired upward gaze often observed in older individuals [29]. The closer proximity of the SR to the globe, combined with weakened connective tissues, restricts its ability to generate effective and rapid eye movements, particularly during upward gaze, a key function of the SR [30].

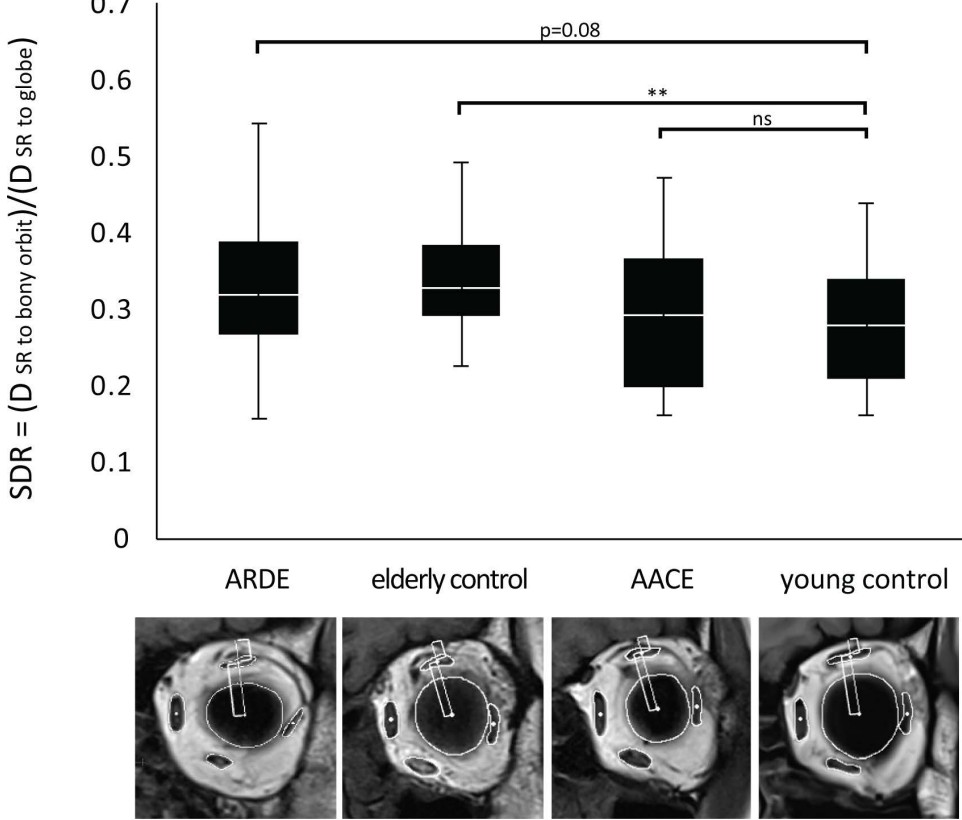

**Fig 5. Box plots displaying the relative positions of the SR across the four study groups, quantified as the SR downward displacement ratio (SDR) between the distance from the SR centroid to the bony orbit, and the distance from the globe centroid to the SR centroid.** The higher SDR observed in both the ARDE group and the elderly control group signifies that the SR is positioned closer to the globe and further from the orbital wall. This indicates a tendency for the SR to shift its position downward with increasing age. **, $p < 0.01$.

A key strength of this study is the comparison of ARDE and AACE, both of which share the characteristic of often presenting with sudden-onset esotropia, analyzed alongside age-matched controls using MRI. When AACE is associated with presbyopia and arises later in life [31], MRI helps differentiate it from ARDE. AACE can often be managed conservatively with reading glasses, topical cycloplegics, and reduced near work, whereas ARDE typically requires surgical intervention, such as MR recession or LR superior transposition, to correct the anatomical abnormalities. In addition, the increased MR-to-LR CSA ratio observed in both esotropia groups may have prognostic value, particularly in AACE. Chronic esotropia can lead to sustained MR overaction and hypertrophy [32], which may reduce responsiveness to conservative treatment. In such cases, an enlarged MR on MRI may reflect a longer-standing deviation with structural remodeling, suggesting a greater role of mechanical factors and highlighting the importance of early, tailored intervention.

In conclusion, this study has revealed substantial LR sagging in ARDE patients relative to older controls, with both esotropia groups exhibiting a higher MR-to-LR CSA ratio, which represents a mechanical factor contributing to esotropia pathology. The SR was also found to be closer to the globe in older subjects, suggesting age-related structural changes. These results highlight the importance of mechanical factors in the pathophysiology of ARDE and AACE, providing insights into the impact of aging on orbital structures and offering guidance for treatment strategies. This study has further demonstrated that MRI is a valuable research tool for exploring the anatomical factors that contribute to the etiology of different forms of strabismus.

## Supporting Information

**S1 Fig. Orbital Imaging Assisted Diagnostic Flowchart for ARDE & AACE.**
(TIF)

**S2 Appendix. Raw measurement data.**
(XLSX)

## Author contributions

**Conceptualization:** Hyun Jin Shin.

**Data curation:** Jin Hee Kim.

**Formal analysis:** Jin Hee Kim.

**Investigation:** Jin Hee Kim.

**Methodology:** Won-Jin Moon, Hyun Jin Shin.

**Project administration:** Hyun Jin Shin.

**Resources:** Won-Jin Moon, Hyun Jin Shin.

**Software:** Jin Hee Kim.

**Supervision:** Hyun Jin Shin.

**Validation:** Hyun Jin Shin.

**Visualization:** Jin Hee Kim.

**Writing – original draft:** Jin Hee Kim, Hyun Jin Shin.

**Writing – review & editing:** Won-Jin Moon, Hyun Jin Shin.

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
