## [Decision Letter · Decision Letter 0]

26 Feb 2025

PONE-D-24-59237Comparison of orbital structures between age-related distance esotropia and acute acquired concomitant esotropia using magnetic resonance imaging and its clinical implicationsPLOS ONE

Dear Dr. Shin,

Thank you for submitting your manuscript to PLOS ONE. After careful consideration, we feel that it has merit but does not fully meet PLOS ONE’s publication criteria as it currently stands. Therefore, we invite you to submit a revised version of the manuscript that addresses the points raised during the review process.

We look forward to receiving your revised manuscript.

Kind regards,

Pawel Klosowski, D.Sc.

Academic Editor

PLOS ONE

Journal Requirements:

“This research was supported by Basic Science Research Program through the National Research Foundation of Korea (NRF) grant funded by the Korea government (MSIT) (No. RS-2023-00251281).”

**Additional Editor Comments:**

Two reviewers still suggest some changes in the paper. Please consider their requirements.

Reviewers' comments:

Reviewer's Responses to Questions

**Comments to the Author**

1. Is the manuscript technically sound, and do the data support the conclusions?

Reviewer #1: Yes

Reviewer #2: Yes

Reviewer #3: Yes

2. Has the statistical analysis been performed appropriately and rigorously? 

Reviewer #1: Yes

Reviewer #2: Yes

Reviewer #3: Yes

3. Have the authors made all data underlying the findings in their manuscript fully available?

Reviewer #1: Yes

Reviewer #2: Yes

Reviewer #3: Yes

4. Is the manuscript presented in an intelligible fashion and written in standard English?

Reviewer #1: Yes

Reviewer #2: Yes

Reviewer #3: Yes

5. Review Comments to the Author

Reviewer #1: Feedback on Manuscript entitled “Comparison of Orbital Structures Between Age-Related Distance Esotropia and Acute Acquired Concomitant Esotropia Using Magnetic Resonance Imaging and Its Clinical Implications”.

The manuscript is well-written and presents an interesting study on the use of MRI in analyzing extraocular muscles in the aged population with strabismus. I recommend acceptance with minor revisions.

Title: Consider shortening the title to better reflect your key findings while maintaining clarity.

Table 1:

The bold formatting is difficult to distinguish; consider using asterisks to denote significant values for clarity.

Standardize the measurement units—currently, some values are in degrees while others are in prism diopters. Choose one format to facilitate comparison.

Supplementary Data Discrepancy:

There is an inconsistency between the number of subjects reported in the abstract (22) and the supplementary data (12). A similar discrepancy appears in the categorization of ARDE, elderly control, and AACE groups. Please clarify and ensure consistency throughout the manuscript.

Revise the affected sentences accordingly to maintain accuracy.

Overall, this is a valuable contribution to an ongoing area of research, and I look forward to seeing the final version.

Reviewer #2: “Comparison of orbital structures between age-related distance esotropia and acute acquired concomitant esotropia using magnetic resonance imaging and its clinical implications”

This manuscript is interesting, adds some new insights on acquired esotropia. The study was carefully conducted using magnetic resonance imaging. Appropriate statistical methods were used and the manuscript is well written, requiring no language editing. The discussion is relevant and well written. However, there are several issues of concern. These are:

1. In Line 251 and Line 308, the sentence, “the SR was positioned closer to the globe”, is difficult to be understood. Was it presumed from the result of “SR downward displacement ratio”? It is necessary to be described how “SR downward displacement ratio” leaded to “the SR was positioned closer to the globe”.

2. The authors cited Gomez et al’s report to support their findings on the closer SR to globe on Line 309. However, description of “closer SR to Globe” could not be found in Gomez, et al, J Optometry, 2018. Also, the description, “This observation also indicates that patients with a higher MR-to-LR CSA ratio might experience less benefit from topical cycloplegics.” Line 302, could not be found in Hayashi, et al, BMC Ophthalmol, 2022. The accuracy of other citations should be confirmed.

Reviewer #3: The study is interesting and novel.

Few points can be added in the manuscript.

1. structured reporting format for MRI reporting in these cases or a flowchart based approach to diagnosis in the discussion section

2. results and discussion are slightly lengthy can can be shortened a bit. Tables can be used in result section.

3. While the results have undeniable diagnostic value, the importance of these parameters in management and prognosis should be stressed upon.

4. Besides the angle and CSA and displacement, is there any signal alteration in the muscles.

6. PLOS authors have the option to publish the peer review history of their article (what does this mean? ). If published, this will include your full peer review and any attached files.

**Do you want your identity to be public for this peer review?** For information about this choice, including consent withdrawal, please see our Privacy Policy .

Reviewer #1: **Yes: ** Suraj Upadhyaya

Reviewer #2: No

Reviewer #3: **Yes: ** Ishan Kumar

---

## [Author Response · Author response to Decision Letter 1]

11 Apr 2025

Reviewer 1

We are deeply grateful for your sincere and valuable comments that have resulted in significant manuscript improvements. We hope that our responses satisfactorily address your concerns.

Question (1)

Title: Consider shortening the title to better reflect your key findings while maintaining clarity.

Author’s reply: Thank you for your feedback. The original title of “Comparison of orbital structures between age-related distance esotropia and acute acquired concomitant esotropia using magnetic resonance imaging and its clinical implications.” has been revised to “Comparison of orbital imaging between age-related distance esotropia and acute acquired concomitant esotropia.”

Question (2)

Table 1: The bold formatting is difficult to distinguish; consider using asterisks to denote significant values for clarity. Standardize the measurement units—currently, some values are in degrees while others are in prism diopters. Choose one format to facilitate comparison.

Author’s reply: Thank you for your thoughtful comments. We have replaced bold formatting with asterisks for clarity. The revised table is shown down below. (Changes made are highlighted in yellow)

Secondly, regarding measurement standardization, we have followed the standard units (Chaudhuri et al., JAMA Ophthalmol. 2013), where angulation of the lateral rectus and the angle between the lateral and superior rectus (rectus muscle displacement angles) are expressed in degrees, while strabismus angles are measured in prism diopters (PD). We appreciate your attention to this detail, which has helped us ensure consistency and accuracy in our reporting.

Question (3)

Supplementary Data Discrepancy:

There is an inconsistency between the number of subjects reported in the abstract (22) and the supplementary data (12). A similar discrepancy appears in the categorization of ARDE, elderly control, and AACE groups. Please clarify and ensure consistency throughout the manuscript. Revise the affected sentences accordingly to maintain accuracy.

Author’s reply: We appreciate your attention to this detail. We have revised and clarified the abstract and supplementary data so that the reported number of orbits aligns with the categorization in the supplementary file.

Thank you again for your insightful questions and valuable advice. We have made every effort to address your inquiries thoroughly in our revised manuscript. We hope our responses effectively resolve any concerns and contribute to a clearer understanding of our study. We are grateful for the opportunity to enhance our work with your feedback.

Reviewer 2

Question (1)

In Line 251 and Line 308, the sentence, “the SR was positioned closer to the globe”, is difficult to be understood. Was it presumed from the result of “SR downward displacement ratio”? It is necessary to be described how “SR downward displacement ratio” led to “the SR was positioned closer to the globe”.

Author’s reply: Thank you for your thoughtful comment.

A larger SR downward displacement ratio (SDR) indicates that closer SR to the globe. This shift is likely due to age-related weakening of orbital connective tissues or the pulleys. In response to your valuable comment, we have clarified this point in both the Methods and Discussion sections to enhance understanding.

Line 146-149: “CSAs of the MR and LR, as well as their ratio, were measured to assess the balance between the two muscles. A higher MR-to-LR CSA ratio indicates that the medial rectus has a relatively larger cross-sectional area compared to the lateral rectus.”

Line 299-300: “In older participants, including ARDE patients and elderly controls, the SR was positioned closer to the globe, as indicated by a significantly larger SDR.”

Question (2)

The authors cited Gomez et al’s report to support their findings on (a) the closer SR to the globe on Line 309. However, description of “closer SR to Globe” could not be found in Gomez, et al, J Optometry, 2018.

Author’s reply: Thank you for your thoughtful comment regarding accuracy of citations.

From line 309 we have described that, “This observation is consistent with Gomez et al. reporting that the SR was geometrically closer to the globe in their ARDE group than younger controls [16].”

In the study by Gomez et al. they reported age-related distance esotropia group exhibited a significantly lower y-axis coordinate of SR compared to the young control group (G.1). This implies that the SR in the ARDE group is vertically positioned closer to the geometric center of the globe than in the younger controls. This is consistent with our finding that SR was geometrically closer to the globe in the ARDE group than in the younger controls.

Question (3)

Also, the description, “This observation also indicates that patients with a higher MR-to-LR CSA ratio might experience less benefit from topical cycloplegics.” Line 302, could not be found in Hayashi, et al, BMC Ophthalmol, 2022. The accuracy of other citations should be confirmed.

Author’s reply: We thank the reviewer for pointing this out. The sentence in Line 302—“This observation also indicates that patients with a higher MR-to-LR CSA ratio might experience less benefit from topical cycloplegics”—was not directly supported by the cited article (Hayashi et al., BMC Ophthalmol, 2022).

While that study reported a negative correlation between esotropia angle reduction and the duration of untreated esotropia, it did not specifically address muscle size or the MR-to-LR CSA ratio.

Our original intent was to hypothesize that chronic esotropia may lead to sustained overaction and hypertrophy of the medial rectus (MR), which could reduce the efficacy of conservative treatments in some AACE cases. This is supported by previous findings showing elevated levels of insulin-like growth factor-1 (IGF-1) in the rectus muscle of esotropia patients, suggesting ongoing muscle remodeling (Hao et al. Alteration of Neurotrophic Factors and Innervation in Extraocular Muscles of Individuals With Concomitant Esotropia. IOVS 2024) In this context, a significantly enlarged MR observed on MRI may suggest a longer-standing deviation accompanied by secondary structural changes, indicating a greater contribution of mechanical rather than neurologic factors. Such cases may show a limited response to conservative treatments, such as cycloplegic eye drops.

However, since our study did not investigate the duration of strabismus or serial changes in MR size, we acknowledge that the aforementioned statement extends beyond the scope of our data. In accordance with your suggestion, we have removed the sentence (This observation also indicates …) in the revised manuscript.

To maintain scientific clarity, we have instead retained and slightly expanded the following sentence:

"Further investigation is required to determine whether the increased MR-to-LR CSA ratio in AACE is associated with a longer duration of strabismus, which would help clarify the relationship between anatomical changes in the EOM and disease chronicity."

Thank you for your valuable feedback.

Thank you again for your insightful questions and valuable advice. We have made every effort to address your inquiries thoroughly in our revised manuscript. We hope our responses effectively resolve any concerns and contribute to a clearer understanding of our study. We are grateful for the opportunity to enhance our work with your feedback.

Reviewer 3

Question (1)

Structured reporting format for MRI reporting in these cases or a flowchart based approach to diagnosis in the discussion section

Orbital Imaging Assisted Diagnostic Flowchart for ARDE & AACE

Author’s reply: Thank you for your valuable feedback on structured orbital imaging reporting and the diagnostic approach. We have incorporated a flowchart in the discussion section as a S1 Figure to provide a clearer and more systematic representation of the diagnostic process.

Question (2)

Results and discussion are slightly lengthy, can be shortened a bit. Tables can be used in the result section.

Thank you for your valuable comment. We appreciate your suggestion and have revised the Results and Discussion sections to improve conciseness while maintaining clarity.

Character count of discussion section was reduced from 1,323 to 1,276. We believe these modifications enhance the readability of the manuscript.

Question (3)

While the results have undeniable diagnostic value, the importance of these parameters in management and prognosis should be stressed upon.

Author’s reply: parameters. We agree that these findings not only provide diagnostic value but also offer meaningful insights into management and prognosis.

First, a key strength of this study lies in the comparative analysis of ARDE and AACE—two entities that often present similarly with sudden-onset esotropia but require distinctly different treatment approaches. MRI can aid in distinguishing these conditions, especially in older patients with AACE associated with presbyopia. In such cases, AACE may initially respond well to conservative treatments, including reading glasses, topical cycloplegics, and reduced near work. In contrast, ARDE typically does not respond to these measures and more often necessitates surgical intervention, such as medial rectus (MR) recession or superior transposition of the lateral rectus (LR), to address the mechanical and anatomical alterations observed on imaging.

Second, the increased MR-to-LR CSA ratio observed in both esotropia groups has potential prognostic implications, particularly in AACE. Chronic esotropia may lead to sustained MR overaction and resultant hypertrophy. Previous studies have also reported elevated levels of insulin-like growth factor-1 (IGF-1) in the lateral rectus muscle of AACE patients, suggesting active muscle remodeling. In this context, a significantly enlarged MR on MRI could indicate a longer-standing deviation with secondary structural changes, implying a greater contribution of mechanical factors. Such cases may exhibit limited response to conservative therapy, underscoring the importance of early intervention and individualized treatment planning.

In response your comment, we added this clinical point in the discussion section as following (line 313-324):

“A key strength of this study is the comparison of ARDE and AACE, both of which share the characteristic of often presenting with sudden-onset esotropia, analyzed alongside age-matched controls using MRI. When AACE is associated with presbyopia and arises later in life [31], MRI helps differentiate it from ARDE. AACE can often be managed conservatively with reading glasses, topical cycloplegics, and reduced near work, whereas ARDE typically requires surgical intervention, such as MR recession or LR superior transposition, to correct the anatomical abnormalities. In addition, the increased MR-to-LR CSA ratio observed in both esotropia groups may have prognostic value, particularly in AACE. Chronic esotropia can lead to sustained MR overaction and hypertrophy [32], which may reduce responsiveness to conservative treatment. In such cases, an enlarged MR on MRI may reflect a longer-standing deviation with structural remodeling, suggesting a greater role of mechanical factors and highlighting the importance of early, tailored intervention.”

Question (4)

Besides the angle and CSA and displacement, is there any signal alteration in the muscles.

We appreciate the reviewer’s insightful question. In this study, our primary focus was on quantitative anatomical parameters derived from orbital MRI. Specifically, we analyzed the following imaging features:

Horizontal rectus (HR) displacement angles – These were measured to assess the vertical displacement of the medial rectus (MR) and lateral rectus (LR) muscles in relation to a defined horizontal reference line. Significant inferior displacement of the LR was observed in the ARDE group compared to elderly controls.

Lateral rectus (LR) tilting angle – This represents the orientation of the LR muscle relative to a vertical reference line. A more oblique (nasal-temporal) tilt was identified in the ARDE group, further supporting the presence of structural changes in this condition.

MR-to-LR cross-sectional area (CSA) ratio – This was calculated to evaluate the relative muscle bulk between the MR and LR. Both the ARDE and AACE groups demonstrated significantly higher MR-to-LR CSA ratios than their respective age-matched controls, suggesting an imbalance in the horizontal rectus muscles.

Superior rectus (SR) downward displacement ratio (SDR) – This ratio was used to estimate the proximity of the SR to the globe, with higher values indicating a closer position. Increased SDRs were observed in both the ARDE group and elderly controls, suggesting age-related displacement patterns.

We did not observe any abnormal signal intensity changes (e.g., hyperintensity) within the extraocular muscles on T1-weighted MRI, which might have indicated muscle inflammation, edema, or fibrosis. Such signal alterations are typically observed in inflammatory orbital diseases, such as thyroid eye disease, rather than in the conditions investigated in our study.

We sincerely appreciate your thoughtful and constructive feedback again. Your comments have been invaluable in enhancing the overall quality of our manuscript.

---

## [Decision Letter · Decision Letter 1]

20 Apr 2025

Comparison of orbital structures between age-related distance esotropia and acute acquired concomitant esotropia using magnetic resonance imaging and its clinical implications

PONE-D-24-59237R1

Dear Dr. Shin,

We’re pleased to inform you that your manuscript has been judged scientifically suitable for publication and will be formally accepted for publication once it meets all outstanding technical requirements.

Kind regards,

Pawel Klosowski, D.Sc.

Academic Editor

PLOS ONE

Additional Editor Comments (optional):

Reviewers' comments:

Reviewer's Responses to Questions

**Comments to the Author**

1. If the authors have adequately addressed your comments raised in a previous round of review and you feel that this manuscript is now acceptable for publication, you may indicate that here to bypass the “Comments to the Author” section, enter your conflict of interest statement in the “Confidential to Editor” section, and submit your "Accept" recommendation.

Reviewer #2: All comments have been addressed

2. Is the manuscript technically sound, and do the data support the conclusions?

Reviewer #2: Yes

3. Has the statistical analysis been performed appropriately and rigorously? 

Reviewer #2: Yes

4. Have the authors made all data underlying the findings in their manuscript fully available?

Reviewer #2: Yes

5. Is the manuscript presented in an intelligible fashion and written in standard English?

Reviewer #2: Yes

6. Review Comments to the Author

Reviewer #2: (No Response)

7. PLOS authors have the option to publish the peer review history of their article (what does this mean? ). If published, this will include your full peer review and any attached files.

**Do you want your identity to be public for this peer review?** For information about this choice, including consent withdrawal, please see our Privacy Policy .

Reviewer #2: No

---

## [Editor Report · Acceptance letter]

PONE-D-24-59237R1

PLOS ONE

Dear Dr. Shin,

I'm pleased to inform you that your manuscript has been deemed suitable for publication in PLOS ONE. Congratulations! Your manuscript is now being handed over to our production team.

Kind regards,

on behalf of

Prof. Pawel Klosowski

Academic Editor

PLOS ONE